# From Interaction to Intervention: How Mesenchymal Stem Cells Affect and Target Triple-Negative Breast Cancer

**DOI:** 10.3390/biomedicines11041182

**Published:** 2023-04-15

**Authors:** Yong Wu, Hang Chee Erin Shum, Ke Wu, Jaydutt Vadgama

**Affiliations:** 1Division of Cancer Research and Training, Department of Internal Medicine, Charles R. Drew University of Medicine and Science, David Geffen UCLA School of Medicine and UCLA Jonsson Comprehensive Cancer Center, Los Angeles, CA 90095, USA; 2Li Ka Shing Faculty of Medicine, The University of Hong Kong, Hong Kong, China

**Keywords:** triple-negative breast cancer, mesenchymal stem cells, tumor microenvironment, proliferation, metastasis

## Abstract

Triple-negative breast cancer (TNBC) lacks estrogen receptor, progesterone receptor, and human epidermal growth factor receptor 2 expressions, making targeted therapies ineffective. Mesenchymal stem cells (MSCs) have emerged as a promising approach for TNBC treatment by modulating the tumor microenvironment (TME) and interacting with cancer cells. This review aims to comprehensively overview the role of MSCs in TNBC treatment, including their mechanisms of action and application strategies. We analyze the interactions between MSC and TNBC cells, including the impact of MSCs on TNBC cell proliferation, migration, invasion, metastasis, angiogenesis, and drug resistance, along with the signaling pathways and molecular mechanisms involved. We also explore the impact of MSCs on other components of the TME, such as immune and stromal cells, and the underlying mechanisms. The review discusses the application strategies of MSCs in TNBC treatment, including their use as cell or drug carriers and the advantages and limitations of different types and sources of MSCs in terms of safety and efficacy. Finally, we discuss the challenges and prospects of MSCs in TNBC treatment and propose potential solutions or improvement methods. Overall, this review provides valuable insights into the potential of MSCs as a novel therapeutic approach for TNBC treatment.

## 1. Introduction

### 1.1. Triple-Negative Breast Cancer: Definition, Epidemiology, Clinical Features, Prognosis, and Treatment Challenges

Triple-negative breast cancer (TNBC) is a specific subtype of breast cancer defined by the absence of estrogen receptor (ER), progesterone receptor (PR), and human epidermal growth factor receptor 2 (HER2) expressions [1]. TNBC has been further subdivided into six distinct molecular subtypes: basal-like 1 (BL1), basal-like 2 (BL2), mesenchymal (M), mesenchymal stem–like (MSL), immunomodulatory (IM), and luminal androgen receptor (LAR) [2]. It constitutes 10–15% of all breast cancers and has a worse prognosis than other subtypes [3,4]. TNBC is more prevalent among African American women, premenopausal women, and individuals carrying BRCA1/2 or PALB2 mutations [5].

TNBC is characterized by high-grade tumors with a high proliferation rate and genomic instability [6]. The clinical and epidemiological behavior of TNBC is heterogeneous due to its biological diversity [3]. Moreover, TNBC is generally unresponsive to hormone therapy or HER2 inhibitors, which are effective for other subtypes [7]. Consequently, chemotherapy remains the primary treatment option, although recent advances show promise with PARP inhibitors for BRCA-mutated patients and immunotherapy for PD-L1-positive patients [3].

The challenges in treating TNBC include identifying biomarkers for predicting responses to therapy, overcoming drug resistance mechanisms, enhancing immune activation, and addressing social-economic disparities that affect access to care [3,5]. Future research should focus on developing innovative agents targeting specific molecular pathways or vulnerabilities in TNBC subtypes and improving risk assessment, screening, and prevention strategies based on genomic and epidemiologic factors.

### 1.2. Sources, Characteristics, Functions, and Application Fields of Mesenchymal Stem Cells (MSCs)

Mesenchymal stem cells (MSCs) are a class of multipotent stromal cells that possess the capacity to differentiate into various cell types, including bone, cartilage, muscle, and adipose cells [8,9]. These cells are obtained from diverse tissues, such as bone marrow, umbilical cord blood, adipose tissue, and dental pulp [10,11]. MSCs possess several advantageous characteristics that render them attractive for biomedical applications, including their immunomodulatory properties, homing ability, capacity to secrete trophic factors, and low immunogenicity.

MSCs exhibit significant potential in numerous fields, such as tissue engineering, regenerative medicine, cell therapy, and gene therapy. These cells have been utilized to treat various ailments, such as osteoarthritis, spinal cord injury, graft-versus-host disease, myocardial infarction, and diabetes mellitus. Nevertheless, MSCs possess a few challenges and constraints, such as variabilities in quality and quantity, a lack of standardized protocols for isolation and expansion, a risk of contamination, and tumorigenicity.

MSCs present a hopeful avenue for clinical applications in the future. However, more extensive research is necessary to comprehend their biology comprehensively and optimize their safety and efficacy. It is critical to establish clear criteria for identifying MSCs based on their molecular markers and functional properties. Additionally, it is imperative to develop reliable in vivo tracking methods for MSCs after transplantation. Furthermore, evaluating the long-term outcomes of MSC-based therapies in large-scale randomized controlled trials is crucial.

### 1.3. The Purpose and Scope of this Review

This review aims to present a comprehensive summary of the current knowledge regarding the role of mesenchymal stem cells (MSCs) in triple-negative breast cancer (TNBC), a subtype of breast cancer that is highly aggressive and heterogeneous and currently lacks effective targeted therapies. We will concentrate on elucidating the mechanisms underlying the interactions between MSCs and TNBC cells, along with other constituents of the tumor microenvironment (TME), and how these interactions impact various aspects of TNBC progression and treatment response. Furthermore, we will examine the potential applications of MSCs as cellular or drug carriers in TNBC therapy, comparing different strategies that aim to improve their therapeutic efficacy and safety. Finally, we will examine the principal challenges and prospects of MSC-based treatment for TNBC and suggest potential avenues for further research and development.

We trust that this review will offer a comprehensive and coherent overview of the role of MSCs in TNBC, stimulating more interest and investigation in this promising area of research. Additionally, we hope our work will inspire new ideas and approaches to enhance the diagnosis, prognosis, and treatment of TNBC patients.

## 2. Effect of MSCs on TNBC Cells and Their Tumor Microenvironment and Its Molecular Mechanism

### 2.1. Context-Dependent Effects of MSC–TNBC Interactions on Proliferation, Migration, Invasion, and Drug Resistance

Direct cellular interaction between mesenchymal stem cells (MSCs) and triple-negative breast cancer (TNBC) cells have been shown to trigger the epithelial-mesenchymal transition (EMT) process in TNBC cells. This transition enhances their invasive and metastatic properties [12,13,14]. In addition, MSCs can transmit mitochondria to TNBC cells using either tunneling nanotubes or extracellular vesicles. This transfer, in turn, promotes oxidative phosphorylation and increases chemoresistance in TNBC cells [15]. Furthermore, MSCs can fuse with cancer cells, creating hybrid cells that possess a heightened tumorigenicity and stemness [16,17].

Mesenchymal stem cells (MSCs) can influence triple-negative breast cancer (TNBC) cells through the indirect secretion of soluble factors. For instance, MSCs can secrete interleukin-6 (IL-6) and prostaglandin E2 (PGE2), which activate the IL-6/STAT3/PGE2 positive feedback loop in TNBC cells [18]. This loop, in turn, promotes tumor growth, angiogenesis, inflammation, immune evasion, epithelial-mesenchymal transition (EMT), stemness, and drug resistance. Additionally, MSCs can secrete transforming growth factor-beta (TGF-β), which stimulates EMT and immunosuppression in TNBC cells [19]. Moreover, MSCs can secrete vascular endothelial growth factor (VEGF), which promotes angiogenesis and enhances tumor perfusion [20].

The interaction between mesenchymal stem cells (MSCs) and triple-negative breast cancer (TNBC) cells is bidirectional and context-dependent, and its effects depend on the molecular characteristics of both MSCs and TNBC cells [18]. For instance, studies have reported that MSCs can promote TNBC cell proliferation, migration, invasion, and drug resistance by secreting factors like interleukin-6 (IL-6) and prostaglandin E2 (PGE2) [18]. Conversely, other studies have demonstrated that MSCs can induce TNBC cell apoptosis or autophagy through the transfection and synthesis of miRNAs, or by inhibiting bone marrow-derived MSCs [18]. In addition, MSCs can express tumor necrosis factor-related apoptosis-inducing ligand (TRAIL), which can trigger apoptosis in TNBC cells [21]. MSCs can also inhibit the expression of cyclin D1, a key regulator of cell cycle progression [22,23,24]. Furthermore, MSCs can sensitize TNBC cells to chemotherapy by increasing the expression of drug transporters or reducing the expression of drug efflux pumps [25,26]. Therefore, the interaction between MSCs and TNBC cells is intricate and diverse (Figure 1), and further research is required to elucidate its underlying mechanisms and impact.

Therefore, comprehending the mechanisms and ramifications underlying the interplay between MSCs and TNBC assumes paramount importance in devising innovative therapeutic modalities for TNBC [13,18]. By targeting pivotal molecules implicated in this interplay (including, but not limited to, interleukin-6 (IL-6), prostaglandin E2 (PGE2), transforming growth factor-beta (TGF-β), vascular endothelial growth factor (VEGF), and fibroblast activation protein alpha (FAPα) [13]), novel prospects for surmounting the challenges posed by this formidable breast cancer subtype may emerge.

### 2.2. Effect of MSCs on Other Components (such as Immune Cells and Stromal Cells) in TNBC Tumor Microenvironment (TME)

MSCs are capable of influencing various components of the tumor microenvironment (TME) in TNBC through direct cell-to-cell contact or by secreting cytokines, chemokines, growth factors, and extracellular vesicles (EVs) [20,27]. These interactions have the potential to modulate the immune response, angiogenesis, inflammation, fibrosis, and metastasis in TNBC [27,28]. For instance, MSCs have been found to recruit and polarize macrophages towards the M2 phenotype, which promotes tumor growth and invasion [29]. Additionally, MSCs can induce regulatory T cells (Tregs) and myeloid-derived suppressor cells (MDSCs), which may result in the suppression of anti-tumor immunity [30]. Furthermore, MSCs can secrete vascular endothelial growth factor (VEGF) that stimulates angiogenesis and enhances tumor oxygenation [31,32,33]. MSCs can also interact with cancer-associated fibroblasts (CAFs), which produce extracellular matrix (ECM) components and matrix metalloproteinases (MMPs) that facilitate tumor invasion and metastasis [18,34,35]. Lastly, MSC-derived EVs are capable of transferring microRNAs (miRNAs), long non-coding RNAs (lncRNAs), or proteins to TNBC cells or other TME cells, which can modulate their gene expression and signaling pathways [36,37,38].

## 3. Application Strategy of MSCs in TNBC Treatment

### 3.1. Advantages and Limitations of MSCs as a Cell Carrier or Drug Carrier in the Treatment of TNBC

MSCs are a group of multipotent cells that can differentiate into various types of cells and exhibit immunomodulatory and anti-inflammatory properties. MSCs have demonstrated an impressive capacity for tumor homing, enabling them to migrate and accumulate within neoplastic tissues following systemic injection [39]. This unique characteristic positions MSCs as promising options for tumor-targeted therapy, particularly for triple-negative breast cancer (TNBC), a subtype of breast cancer with high aggressiveness and heterogeneity and limited treatment alternatives [40]. The potential of MSCs as tumor-targeting carriers presents a significant opportunity for developing effective therapies for TNBC and other cancer types.

Numerous studies have delved into the utility of mesenchymal stem cells (MSCs) as cell or drug carriers in treating TNBC. These multipotent cells can be genetically engineered to express anticancer genes or proteins, such as interferon-β, tumor necrosis factor-related apoptosis-inducing ligand (TRAIL), interleukin-12, etc., and effectively deliver them to the tumor site [41]. Alternatively, MSCs can be loaded with various anticancer drugs, such as doxorubicin, paclitaxel, gemcitabine, and others, and then they undergo controlled release [42].

Another potential application of MSCs in TNBC treatment is using MSC-derived extracellular vesicles (MSC–EVs) as drug delivery systems. MSCs are capable of secreting EVs or exosomes that contain therapeutic molecules, such as microRNAs, long non-coding RNAs, proteins, and others, which can be transferred to the tumor cells [43]. MSC–EVs are nano-sized membrane-bound vesicles that contain various bioactive molecules, such as proteins, lipids, mRNAs, non-coding RNAs, and DNA fragments [43]. MSC–EVs can mimic the immunomodulatory and anti-inflammatory properties of their parental cells and can also transfer functional cargos to recipient cells [43]. Moreover, MSC–EVs can be engineered to express specific molecules or load specific drugs that can target TNBC cells or modulate the tumor microenvironment [43]. Several studies have reported the use of MSC–EVs as drug delivery systems for TNBC treatment. For example, Chang et al. [44] showed that MSC–EVs loaded with miR-125b inhibited TNBC growth and metastasis by downregulating HIF1α and its target genes. Similarly, Dong et al. [36] demonstrated that MSC–EVs loaded with doxorubicin-induced apoptosis and autophagy in TNBC cells by activating AMPK/mTOR signaling. Furthermore, EVs loaded with therapeutic components such as tumor-suppressor drugs, siRNAs, proteins, peptides, and conjugates exhibit significantly enhanced anti-tumor effects [45]. Chemotherapy drugs are known to harm cancer cells but can also damage other fast-growing cells in the body, leading to side effects such as fatigue. Therefore, using MSC–EVs as drug delivery vehicles may offer a more targeted approach to cancer treatment, potentially reducing side effects. Collectively, these findings suggest that MSC-based therapies hold considerable promise as a novel and practical approach for TNBC treatment.

In spite of the potential benefits, there are some challenges and limitations to consider when using MSCs as cell or drug carriers for TNBC treatment. One of the primary issues is the safety and efficacy of different sources or types of MSCs. MSCs can be derived from various tissues, such as bone marrow (BM–MSCs), adipose tissue (AD–MSCs), umbilical cord blood (UCB–MSCs), and others, and each source may have distinct characteristics and functions [10]. Additionally, MSCs can be either autologous (from the same patient) or allogeneic (from a different donor), which can impact their immunogenicity and compatibility [46,47]. Furthermore, MSCs can be either primary (isolated directly from tissues) or induced (reprogrammed from other cell types), which can affect their differentiation potential and stability [48]. These factors must be considered when selecting and using MSCs as carriers for TNBC therapy.

Therefore, it is critical to thoroughly compare and evaluate the benefits and drawbacks of various sources or types of MSCs with respect to their safety and efficacy in TNBC therapy. Several factors must be taken into consideration, including the accessibility and availability of MSC sources, the quality and quantity of MSC isolation, the conditions for MSC expansion and culture, the methods of genetic modification of MSCs, the loading capacity and release kinetics of drugs on MSCs, the specificity and homing efficiency of MSCs towards TNBC, as well as the immunological response and toxicity of MSCs. It is also necessary to investigate the survival rate and fate of MSCs in vivo, as well as the molecular mechanisms that govern their interactions with TNBC cells. To optimize these parameters, it is crucial to conduct further research and develop standardized protocols for utilizing MSCs as drug or cell carriers for TNBC treatment, while also thoroughly examining any potential adverse effects or risks that may be associated with this approach.

### 3.2. The Effects of Different Methods on the Therapeutic Effect and Safety of MSCs, and the Advantages and Disadvantages of Different Methods in Targeting, Stability, and Controllability

MSCs possess a unique multipotent differentiation ability and immunomodulatory function, making them a popular choice in regenerative medicine and tissue engineering. However, the success of MSC transplantation in vivo is hindered by several challenges, such as low survival rates, lack of specific targeting, senescence, and immune rejection. To address these limitations and enhance the therapeutic effect and safety of MSCs, researchers have employed various modification methods, including gene modification, surface modification, and pretreatment.

Gene modification is a technique that involves using a vector to introduce specific gene fragments into mesenchymal stem cells (MSCs). The primary objective of gene modification is to enhance cellular survival, migration, homing, and adhesion to target sites while also preventing poor MSC division and growth (senescence). Several genes, such as CXC chemokine receptors 1, 4, and 7 [49], Sox2 and Oct4 [50], Bcl-2 [51], and IL-10 [52], have been overexpressed in MSCs to increase their stemness, proliferation, differentiation, anti-inflammatory, and anti-apoptotic effects. The key advantages of gene modification are the ability to confer a stable and long-lasting expression of desired genes in MSCs and introduce multiple genes simultaneously. However, gene modification also poses potential drawbacks, including unwanted side effects such as insertional mutagenesis, immune response, or tumorigenesis [53,54]. Therefore, the careful selection of vectors and genes is critical to ensuring the safety and efficacy of gene-modified MSCs.

Surface modification involves altering the membrane properties of MSCs by attaching specific molecules or particles. The primary goal of surface modification is to enhance the targeting, stability, and controllability of MSCs. Antibodies [55], nanoparticles [55], peptides [56], and aptamers [56] are among the surface modifiers that have been used to improve MSCs’ homing, imaging, drug delivery, and gene editing capabilities. The advantages of surface modification are that it can provide versatile functions for MSCs and avoid the risks associated with genetic manipulation. However, surface modification can also negatively affect MSCs’ viability, differentiation, and immunogenicity [55,57,58]. Therefore, it is essential to carefully select modifiers and methods to ensure the safety and efficacy of surface-modified MSCs.

Pretreatment involves subjecting MSCs to specific stimuli or conditions prior to transplantation. The primary aim of pretreatment is to augment the survival, engraftment, and functionality of MSCs. Pretreatment methods may include hypoxia, heat shock, cytokines, and pharmacological agents [59], leading to the upregulation of anti-inflammatory, anti-apoptotic, and pro-angiogenic factors in MSCs. The advantages of pretreatment are that it can replicate the physiological environment of MSCs and modulate their behavior without genetic or surface modifications. The disadvantages are that it may result in MSCs’ senescence, a loss of differentiation potential, or undesired immune reactions [53,59,60]. Therefore, carefully selecting stimuli and conditions is crucial for the safety and effectiveness of pretreated MSCs.

In summary, the diverse modification techniques exert distinct effects on the therapeutic potential and safety of MSCs (Figure 2). Gene modification can effectively enhance MSCs’ stemness and function, although the approach carries the risk of introducing genetic instability or tumorigenicity. Surface modification can enhance MSCs’ targeting and controllability but can compromise their viability and immunogenicity. Pretreatment can improve MSCs’ survival and engraftment, but it can also lead to MSCs’ senescence or loss of differentiation potential. The advantages and disadvantages of each method must be carefully evaluated based on the specific application and disease context. Future investigations are necessary to optimize the modification approaches and determine their long-term efficacy in vivo.

### 3.3. Effects of Different Routes of Administration on the Efficacy and Safety of MSCs Treatment

MSCs have been extensively studied for their therapeutic potential in various diseases [61,62]. Nevertheless, the delivery route of MSCs is a crucial factor that influences their efficacy and safety. Different delivery routes exhibit varying advantages and disadvantages concerning distribution range, bioavailability, and side effects.

Intravenous injection (IV) represents the most common delivery route for MSCs, primarily due to its convenience and minimally invasive nature. Nonetheless, this delivery method is associated with several drawbacks, including the rapid clearance of MSCs by the immune system and organs such as the lungs and spleen [63,64], the possibility of MSC-induced pulmonary embolism or microvascular occlusion [61], and the low homing efficiency of MSCs to target tissues [63]. Therefore, it is necessary to explore alternative delivery routes to improve the efficacy and safety of MSC-based therapies in clinical applications.

Another potential delivery route for MSCs is a local injection (LI), which involves directly injecting MSCs into the target tissue or organ. LI offers several advantages over IV injection, including higher retention and engraftment rates of MSCs in the target tissue [61], local paracrine effects, and modulation of the microenvironment [61], and the avoidance of systemic side effects such as immunogenicity or tumorigenicity [61]. However, LI also poses certain limitations. Firstly, LI may cause tissue damage or inflammation at the injection site [61]. Secondly, LI may require multiple injections or catheterization for specific organs, such as the heart or brain [61]. Lastly, LI may not be suitable for diffuse diseases that affect multiple organs or tissues [61]. Overall, while LI may be a promising alternative to IV injection, its potential drawbacks must be taken into consideration when selecting an appropriate delivery route for MSC-based therapies in clinical settings.

Apart from IV and LI, other possible delivery routes for MSCs include intra-arterial injection (IA), intraperitoneal injection (IP), intramuscular injection (IM), subcutaneous injection (SC), and intrathecal injection (IT), among others. Each of these routes has its own advantages and disadvantages depending on the disease model and desired therapeutic outcome. Therefore, there is no universal delivery route for MSCs in treating different diseases. The optimal delivery route for MSC-based therapies should be determined by carefully considering various factors, such as the cell dose, cell type, disease stage, target organ/tissue location, size, functionality, vascularization, inflammation status, and other relevant clinical factors [65]. By evaluating these factors, researchers and clinicians can identify the most appropriate delivery route for MSCs that offers the best therapeutic effect and minimizes the risk of adverse events. Therefore, a personalized approach to selecting the optimal delivery route for MSC-based therapies is crucial for achieving optimal clinical outcomes.

In summary, the different delivery routes for MSCs offer varying distribution range, bioavailability, and potential side effects. IV injection is a convenient and minimally invasive method but has low retention and homing efficiency. Local injection, on the other hand, provides higher retention and engraftment rates but may cause tissue damage or inflammation. Other delivery routes have their own unique advantages and disadvantages that depend on several factors. Thus, selecting the appropriate delivery route for MSC-based therapies requires careful consideration of the specific disease, target organ, and patient characteristics. By weighing the benefits and drawbacks of each delivery route and considering individual clinical factors, researchers and clinicians can identify the optimal delivery route that offers the best therapeutic effect with minimal adverse events.

## 4. Challenges and Future Prospects of MSCs in TNBC Treatment

### 4.1. The Main Problems Encountered by MSCs in TNBC Treatment and the Methods to Solve These Problems

Mesenchymal stem cells (MSCs) have been extensively investigated as a plausible therapeutic intervention for multiple diseases, including cancer, owing to their remarkable capacity to migrate to the tumor microenvironment and regulate immune responses [66]. The summary of the role of MSCs in the treatment of TNBC has been presented in Table 1. Despite these beneficial attributes, MSCs present several challenges and constraints in treating TNBC, including the risk of transplant rejection, the potential for tumor promotion, and the challenges in controlling drug release, among others (Figure 3).

Transplant rejection is a significant challenge encountered by MSCs in the treatment of TNBC. Typically derived from allogeneic sources such as bone marrow or adipose tissue, MSCs possess low immunogenicity and have demonstrated the ability to suppress immune reactions both in vitro and in vivo [67]. However, repeated injections or prolonged exposure to MSCs may still elicit immune responses from the host, leading to diminished MSC survival and function at the tumor site and unfavorable effects on the host’s health. Various strategies have been proposed to address this issue, including the use of autologous MSCs, genetic engineering of MSCs to express immunomodulatory molecules, or the encapsulation of MSCs with biomaterials [68]. For example, previous studies [69,70,71] have demonstrated the efficacy of engineering human umbilical cord-derived MSCs (hUC-MSCs) to overexpress indoleamine 2, 3-dioxygenase 1 (IDO1), an enzyme that inhibits T cell proliferation by degrading tryptophan. Similarly, Pan et al. [72] coated human adipose-derived stem cells (hASCs) with polyethylene glycol (PEG)-conjugated phospholipid micelles to mask their surface antigens and decrease recognition by NK cells. They observed that PEGylated hASCs exhibited improved survival and homing ability in a mouse model of TNBC compared to uncoated hASCs.

Tumor promotion is another obstacle faced by MSCs in TNBC treatment. MSCs possess a predilection for tumor tissues and can transform into tumor-associated mesenchymal stromal cells (TA–MSCs) that facilitate TNBC metastasis by interacting with tumor-associated macrophages (TAMs) [13]. Furthermore, MSCs can release pro-inflammatory and pro-angiogenic factors that promote tumor growth and invasion [73,74]. As a result, strategies that target the pro-tumor effects of MSCs are essential, including those that aim to inhibit specific signaling pathways or molecules involved in MSC–tumor interactions. Several studies have reported different methods to inhibit the pro-tumor effects of MSCs in TNBC treatment. For instance, Li et al. [13] demonstrated that bone marrow-derived mesenchymal stem cells (BM–MSCs) can be converted into tumor-associated mesenchymal stromal cells (TA–MSCs) to facilitate TNBC metastasis. TA–MSCs secrete multiple C–C motif chemokine ligands that promote CCR2+ tumor-associated macrophage (TAM) recruitment. FAPα is overexpressed in TA–MSCs, which prompts CCR2+ TAM recruitment and polarization. Fibroblast activation protein alpha (FAPα) is crucial in mediating TA–MSC-induced TNBC metastasis. They demonstrated that targeting TA–MSCs with an FAPα-activated prodrug is a promising strategy for suppressing TNBC metastasis. In another study, Ryan et al. [18] silenced IRIS/PKD1 expression in TNBC cells and BM–MSCs using small interfering RNA (siRNA). They revealed that IRIS/PKD1 silencing disrupted the IL-6/PGE 2-positive feedback loop between TNBC cells and BM–MSCs and reduced tumor aggressiveness in vitro and in vivo.

A third challenge faced by MSCs in TNBC treatment is the control of the drug release. While MSCs can selectively target tumors and release drugs locally, regulating the timing and amount of drug release can be difficult. Insufficient or excessive drug release can affect therapeutic efficacy or cause toxicity to normal tissues. Furthermore, the amount of drug release can be influenced by various factors, such as environmental cues (e.g., pH, oxygen level, enzymes), cell viability, and cell differentiation state [75,76]. Thus, it is crucial to develop techniques for regulating drug release from MSCs to meet specific needs. Some possible methods include using stimuli-responsive materials [77], genetic switches [78], or feedback mechanisms [79] to control drug release from MSCs.

MSC-based therapies show promise for enhancing the prognosis of TNBC patients, but further investigation is required to overcome their limitations and refine their parameters. Personalized medicine should be a key consideration in the design and implementation of MSC-based therapies for TNBC patients. This necessitates the assessment of each patient’s unique attributes, including genetic makeup, immune response, and tumor microenvironment, in order to select and tailor the most appropriate type of MSC-based therapy. In doing so, the advantages of MSC-based treatment can be maximized, while the associated risks are minimized.

### 4.2. The Interaction Mechanism between the M Subtype or MSL Subtype of TNBC and MSCs and the Possible Experimental Design Ideas in These Fields

Subsequently, using MSCs to treat TNBC demands further exploration and verification, especially in areas where experimental data support is scarce. Of particular interest are the interaction mechanisms between M subtypes or MSL subtypes and MSCs, which require further investigation. In this regard, we will propose potential experimental designs that could be employed to elucidate these areas of interest.

The M and MSL subtypes of TNBC are identified by elevated gene expression related to epithelial-mesenchymal transition (EMT), angiogenesis, stemness, and remodeling of the extracellular matrix. These subtypes have been linked to unfavorable prognoses, heightened potential for metastasis, and resistance to chemotherapy, as noted in multiple studies [2,80,81].

The interaction between MSCs and M or MSL subtype TNBC cells may involve various mechanisms, including direct cell-to-cell contact, paracrine signaling, extracellular vesicles, and the modulation of the tumor microenvironment. MSCs could stimulate the EMT, angiogenesis, invasion, and stemness of M or MSL subtype TNBC cells by secreting different cytokines, chemokines, growth factors, and matrix metalloproteinases. Moreover, by modulating the expression of immune checkpoint molecules, drug transporters, and apoptosis regulators, MSCs might amplify the drug resistance and immunosuppression of M or MSL subtype TNBC cells, as highlighted in several studies [2,80,81].

To further understand the interaction mechanism between MSCs and M or MSL subtype TNBC cells, several experiments could be conducted. Firstly, MSCs and M or MSL subtype TNBC cells could be co-cultured in vitro, and their gene expression, protein secretion, cell morphology, proliferation, migration, invasion, and stemness could be analyzed using various molecular and cellular techniques such as qRT-PCR, ELISA, immunofluorescence, immunoblotting, flow cytometry, cell counting, wound healing assay, transwell assay, and sphere formation assay [81]. Secondly, MSCs and M or MSL subtype TNBC cells could be injected into immunocompromised mice, and their tumor growth, metastasis, angiogenesis, and immune infiltration could be monitored using various in vivo techniques such as bioluminescence imaging, histology, immunohistochemistry, and flow cytometry [81]. Thirdly, MSCs and M, or MSL-subtype TNBC cells, could be treated with different drugs, such as chemotherapy agents, anti-angiogenic agents, anti-inflammatory agents, or immunotherapy agents, and their drug response, drug resistance, and drug synergy could be evaluated using various in vitro and in vivo techniques such as cell viability assay, apoptosis assay, drug combination index, and tumor regression analysis [81]. Lastly, key molecules or pathways involved in the interaction between MSCs and M, or MSL-subtype TNBC cells, could be identified using various molecular and bioinformatic techniques, such as gene knockdown, gene overexpression, gene editing, microarray, RNA-seq, proteomics, and network analysis [81]. These experiments could potentially shed new light on the molecular and cellular mechanisms of MSCs and M, or MSL-subtype TNBC cell interaction. They may reveal new therapeutic targets or strategies for treating this aggressive breast cancer subtype.

### 4.3. Future Directions and Perspectives of MSCs in TNBC Treatment

MSCs have demonstrated significant potential as a novel therapeutic approach for treating TNBC. However, various challenges and uncertainties must be addressed before their translation into clinical practice. Future directions and perspectives of MSCs in TNBC treatment include:(a)Developing more dependable and standardized methods for MSC isolation, characterization, expansion, modification, delivery, and tracking in vivo, as it would improve the quality control, reproducibility, safety, and efficacy of MSC-based therapies for TNBC.(b)Furthermore, effective ways to enhance the tumor-homing ability and specificity of MSCs, such as utilizing biomimetic materials or surface ligands that can recognize tumor-specific receptors or antigens, need to be explored.(c)Optimizing the timing, dosage, frequency, and combination of MSC administration with conventional or targeted therapies for TNBC is another area of research that requires attention. This would maximize the therapeutic benefits and minimize the adverse effects of MSC-based therapies for TNBC.(d)Investigating the long-term outcomes and potential risks of MSC-based therapies for TNBC, such as tumor recurrence, metastasis, immune rejection, or malignant transformation, is crucial. Achieving this necessitates additional preclinical studies utilizing relevant animal models and clinical trials with adequate follow-up periods.(e)Understanding the molecular mechanisms underlying the interactions between MSCs and TNBC cells or other components of the tumor microenvironment (TME), such as cancer stem cells (CSCs), epithelial-mesenchymal transition (EMT), immune cells, extracellular matrix (ECM), etc., is another area of research that necessitates investigation. This would unveil novel targets and pathways for modulating MSC functions or improving their therapeutic effects for TNBC.

To summarize, MSCs represent a promising avenue for developing personalized and multifaceted therapies for TNBC that could potentially address the limitations of existing treatments. Nonetheless, additional research is essential to unravel the intricate roles of MSCs in TNBC biology and therapy. Furthermore, ethical and legal concerns regarding MSC application must be considered and resolved before clinical implementation.
biomedicines-11-01182-t001_Table 1Table 1Recent studies on the role of mesenchymal stem cells (MSCs) in the treatment of triple-negative breast cancer (TNBC).Research TopicMSC SourceEffectClinical Significance of MSCs in TNBCStudy Author, YearFAPα-expressing TA-MSCBM-MSC/TA- MSCTA-MSCs facilitate TNBC metastasis by interacting with tumor-associated macrophages FAPα-activated prodrug induces FAPα^+^ TA-MSC apoptosis.TA-MSC is a potential target for TNBC anti-metastasis therapy.Li, X. et al., 2021 [13]MSC in BRCA1-IRIS-overexpressing TNBC cells/IL-6/PGE_2_-positive feedback loop between IRISOE TNBC and MSCs increases tumor aggressivenessMSC is a potential therapeutic target in BRCA1-IRIS-overexpressing TNBC.Ryan, D. et al., 2019 [18]TRAIL-expressing MSC with curcumin-loaded chitosan nanoparticles Placental-derived MSCInduces apoptosis in tumor cells Inhibits tumor growth in vivoMSC is an effective anti-TNBC drug carrier.Kamalabadi-Farahani, M. et al., 2018 [82]hAD-MSCs and chemoresistance in TNBChAD-MSChAD-MSCs downregulated miR-106a in TNBC, upregulated ABCG2 and cause doxorubicin resistance.hAD-MSC is a potential therapeutic target to improve chemoresistance in TNBCYeh, W.-L. et al., 2017 [26]Plasmonic-magnetic hybrid nanoparticle (lipids, doxorubicin, gold nanorods, iron oxide nanocluster) loaded MSCshUC-MSCInhibits tumor growth in vivo and in vitroImprove homing ability in IV injectionMSC loaded with nanoparticles act as a novel multifunctional approach for imaging and treatment of TNBC.Xu, C. et al., 2018 [83],Conditioned Medium of MSC Loaded with Paclitaxel AD-MSCStronger inhibitory effects on survival, migration and tumorigenicity for MSC-Paclitaxel conditioned medium than for control and free Paclitaxel in TNBC cell linesMSC is an effective carrier of Paclitaxel. Cordani, N. et al., 2023 [84]Paclitaxel loaded MSC-Derived Exosome Mimetics BM-MSCTherapeutically efficient for TNBC treatment in vitro and *in vivo*MSC-derived exosome mimetic is an effective carrier of Paclitaxel.Kalimuthu, S. et al., 2018 [85]Cannabidiol loaded EVs sensitize TNBC to doxorubicin in both in-vitro and in vivo modelsUC-MSCDecrease side effects Increase therapeutic efficacy of doxorubicin Combination therapy of cannabidiol loaded EV and doxorubicin improves drug safety and efficiency. Patel, N. et al., 2021 [86]Biomimetic nanovesicles made from iPS cell-derived MSC iPS- MSCSuperior cytotoxic effects on doxorubicin resistant TNBC No immunogenicity or toxicityiPSC-MSC nanovesicles are effective doxorubicin carrier.Zhao, Q. et al., 2020 [87]Delivery of miR-381-3p Mimic by MSC-Derived Exosomes AD-MSCInhibits proliferation, migration, and invasion capacity of MDA-MB-231 Promotes apoptosis in vitroMSC-derived exosome as nanocarrier for RNA-based treatmentShojaei, S. et al., 2021 [88]Exosomal delivery of 7SK lnc-RNA hUC-MSCReduced viabilityAltered transcription levels of apoptosis-regulating genesReduced proliferationReduced migration and invasionAltered transcription of EMT-regulating genesReduced in vivo tumor formation abilityMSC-derived exosome is an effective carrier for lncRNA.Farhadi, S. et al., 2023 [89]hUC MSC-derived exosomes loaded with miR-3182 hUC- MSCAbolished cell proliferation and migrationInduced apoptosis in TNBC cells by downregulating mTOR and S6KB1 genesMSC-derived exosome is an effective carrier for microRNA.Khazaei-Poul, Y. et al., 2021 [90]LncRNA HAND2-AS1 influence on MSCs derived exosomal miR-106a-5pBM MSCExosomal-miR-106a-5p secreted by MSCs promoted tumor progression in TNBC cellsMSC derived exosome is a potential therapeutic target in TNBC.Xing, L. et al., 2020 [91]ABCG2; ATP-binding cassette subfamily G member 2; AD-MSC, adipose derived mesenchymal stem cell; BM-MSC, bone marrow mesenchymal stem cell; EMT, epithelial-mesenchymal transition; EV, extracellular vesicles; FAPα, fibroblast activation protein alpha; hAD-MSC, human adipose derived mesenchymal stem cell; hUC- MSC, human umbilical cord mesenchymal stem cell; IL-6/PGE_2_, interleukin-6/prostaglandin E2; iPSC-MSC, induced pluripotent stem cells mesenchymal stem cell; IRISOE, IRIS-overexpressing; IV, intravenous; lncRNA, long noncoding RNA; miR, microRNA; MSC, mesenchymal stem cell; mTOR, mammalian target of rapamycin; TA-MSCs, tumor-associated mesenchymal stromal cell; TNBC, triple negative breast cancer; TRAIL, tumour necrosis factor-related apoptosis inducing ligand.

## Figures and Tables

**Figure 1 biomedicines-11-01182-f001:**
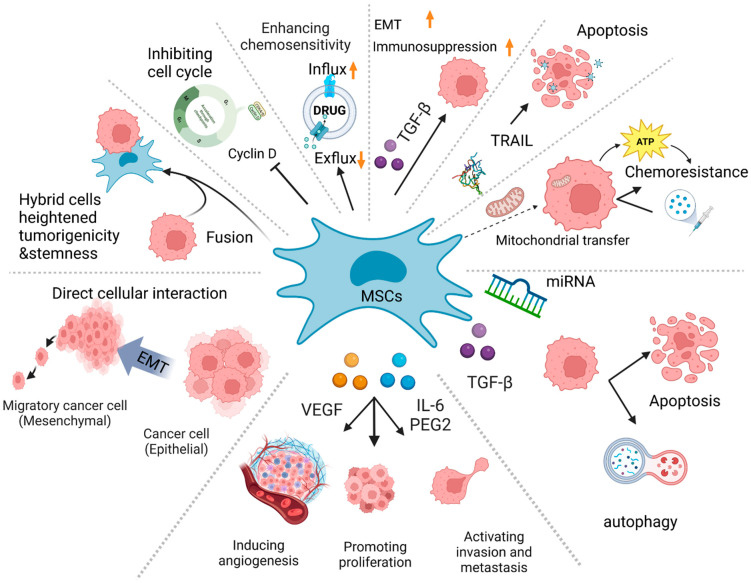
This schematic diagram illustrates how mesenchymal stem cells (MSCs) can interact with triple-negative breast cancer (TNBC) cells, such as through the secretion of soluble factors, transfer of mitochondria, and fusion. These interactions can influence the proliferation, migration, invasion, metastasis, angiogenesis, and drug resistance of TNBC cells. Furthermore, they can also activate signal pathways and molecular mechanisms that could be targeted for therapeutic interventions.

**Figure 2 biomedicines-11-01182-f002:**
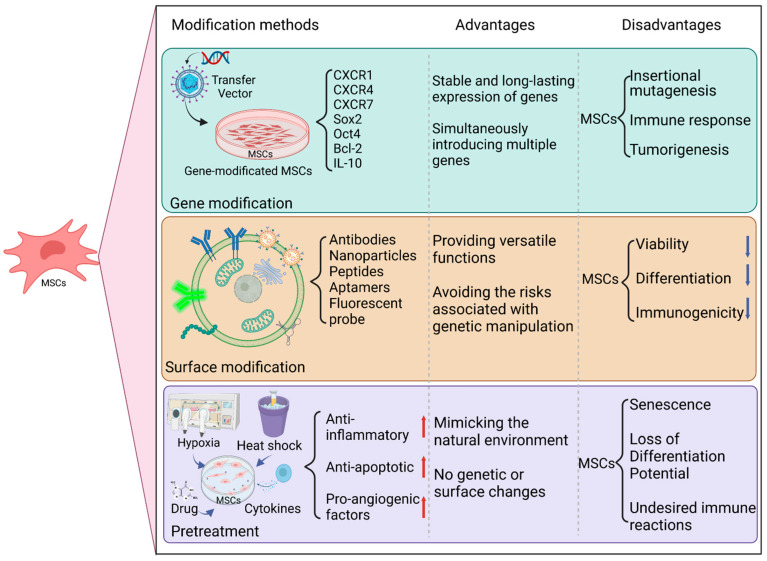
The various methods employed to modify mesenchymal stem cells (MSCs) for therapeutic applications, such as gene modification, surface modification, and pretreatment. Each method can affect the therapeutic effect and safety of MSCs differently, and the advantages and disadvantages of each must be evaluated based on the specific application and disease context.

**Figure 3 biomedicines-11-01182-f003:**
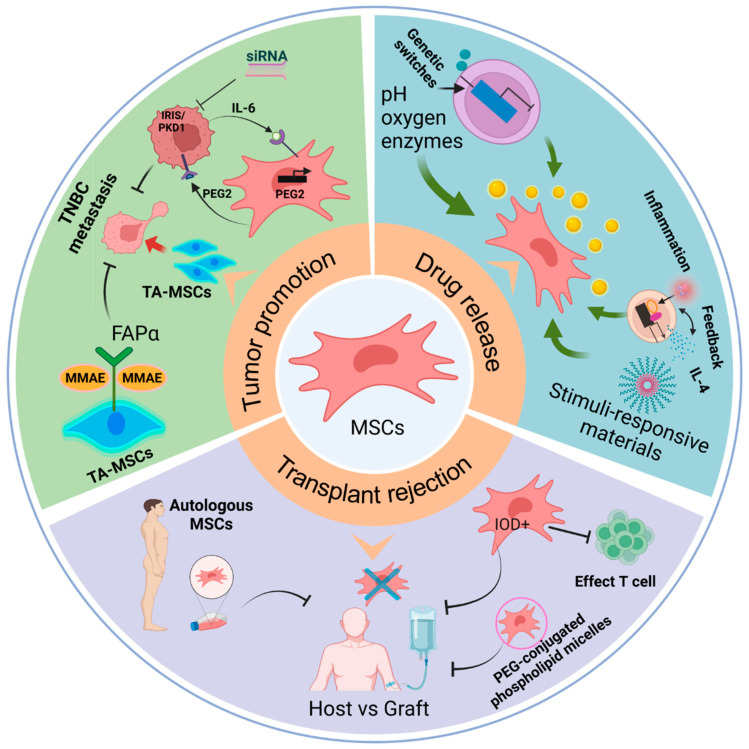
The challenges and proposed solutions for using mesenchymal stem cells (MSCs) in the treatment of TNBC, including transplant rejection, tumor promotion, and drug release control. Strategies to address these challenges include using autologous MSCs, genetic engineering to express immunomodulatory molecules, encapsulation of MSCs with biomaterials, targeting of tumor-associated mesenchymal stromal cells (TA-MSCs) with antibody-drug conjugates, and silencing of signaling pathways involved in MSC-tumor interactions. These methods may be further refined through personalized medicine approaches that consider each patient’s unique attributes.

## Data Availability

Not applicable.

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
