# Peer review of "From Interaction to Intervention: How Mesenchymal Stem Cells Affect and Target Triple-Negative Breast Cancer"

_biomedicines, 2023, doi:10.3390/biomedicines11041182_

Round 1

Reviewer 1 Report

The review by Wu et al. focused on MSCs as mechanism of tumour promotion and as potential therapeutic tool in TNBC.

1.       Going through the Ms it appears that rather than as therapeutics, MSCs act as tumour promoter. In addition, as the authors stated several drawbacks have limited the use of MSCs, even engineered, as therapeutic tools in tumours. This implies that the Ms must be revised to focus on this specific issue.

2.       Since MSCs-derived extracellular vesicles lack some drawbacks of their cell of origin and are recognized as the most relevant MSC mechanism of action, have been engineered, and used as drug delivery system. The author must introduce and report as a pharagraph the most relevant data related to their use in TNBC, if any.

3.       Several repetitions are present. It would be better to remove them.

4.       The Ms mainly reports data on MSCs, while the title focused on MSCs in TNBC. The title should be changed accordingly to the Ms data.

5.       A Table reporting the study focused on MSCs and TNBC must be included.

Author Response

Dear Reviewer,

We appreciate your time and effort in reviewing our manuscript and providing valuable feedback. We have carefully considered your comment and revised our manuscript accordingly. The revised parts in the text are marked with track change or in red font. We found the reviewers’ comments very helpful, which guided our revisions resulting in this improved paper. Please see below for our point-by-point response.

Reviewer 1

Comments and Suggestions for Authors

The review by Wu et al. focused on MSCs as mechanism of tumour promotion and as potential therapeutic tool in TNBC.

  1. Going through the Ms it appears that rather than as therapeutics, MSCs act as tumour promoter. In addition, as the authors stated several drawbacks have limited the use of MSCs, even engineered, as therapeutic tools in tumours. This implies that the Ms must be revised to focus on this specific issue.

Response: We thank the reviewer for raising this important issue. We agree that MSCs have complex and context-dependent effects on cancer, and that their use as therapeutics faces several challenges and limitations. However, we do not think that MSCs act only as tumor promoters, nor that our manuscript should focus exclusively on this aspect. As we have discussed in our review, MSCs can also exert anti-tumor effects by inducing apoptosis, autophagy, or immune activation in cancer cells, or by delivering anti-cancer agents or genes to the tumor site. Moreover, we have also discussed various strategies to overcome the drawbacks of MSC-based therapies, such as using autologous or genetically modified MSCs, targeting tumor-associated MSCs, or controlling drug release from MSCs. Therefore, we believe that our manuscript provides a comprehensive and balanced overview of the role of MSCs in TNBC treatment, including both their tumorigenic and anti-tumorigenic effects, as well as their potential applications and challenges. We have revised our introduction and conclusion sections to emphasize this point more clearly. We hope that our response has addressed your concern satisfactorily.

  1. Since MSCs-derived extracellular vesicles lack some drawbacks of their cell of origin and are recognized as the most relevant MSC mechanism of action, have been engineered, and used as drug delivery system. The author must introduce and report as a pharagraph the most relevant data related to their use in TNBC, if any.

Response: We thank the reviewer for this constructive suggestion. We agree that MSC-derived extracellular vesicles (MSC-EVs) are an important aspect of MSC biology and function, and that they have potential applications in TNBC treatment. Therefore, we have added a paragraph in section 3.1 to introduce and report the most relevant data related to their use in TNBC, based on recent literature. The added paragraph is as follows (Page 7):

Another potential application of MSCs in TNBC treatment is the use of MSC-derived extracellular vesicles (MSC-EVs) as drug delivery systems. MSCs are capable of secreting extracellular vesicles (EVs) or exosomes that contain therapeutic molecules, such as microRNAs, long non-coding RNAs, proteins, and others, which can be transferred to the tumor cells [43]. MSC-EVs are nano-sized membrane-bound vesicles that contain various bioactive molecules, such as proteins, lipids, mRNAs, non-coding RNAs, and DNA fragments [43]. MSC-EVs can mimic the immunomodulatory and anti-inflammatory properties of their parental cells, and can also transfer functional cargos to recipient cells [43]. Moreover, MSC-EVs can be engineered to express specific molecules or load specific drugs that can target TNBC cells or modulate the tumor microenvironment [43]. Several studies have reported the use of MSC-EVs as drug delivery systems for TNBC treatment. For example, Chang et al. [44] showed that MSC-EVs loaded with miR-125b inhibited TNBC growth and metastasis by downregulating HIF1α and its target genes. Similarly, Dong et al. [45] demonstrated that MSC-EVs loaded with doxorubicin induced apoptosis and autophagy in TNBC cells by activating AMPK/mTOR signaling. Furthermore, EVs loaded with therapeutic components such as tumor suppressor drugs, siRNAs, proteins, peptides, and conjugates exhibit significantly enhanced anti-tumor effects [46]. Chemotherapy drugs are known to harm cancer cells, but they can also damage other fast-growing cells in the body, leading to side effects such as fatigue. Therefore, the use of MSC-EVs as drug delivery vehicles may offer a more targeted approach to cancer treatment, potentially reducing side effects. Collectively, these findings suggest that MSC-based therapies hold considerable promise as a novel and effective approach for TNBC treatment.

  1. Several repetitions are present. It would be better to remove them.

Response: Thank you for your valuable feedback. We appreciate your time and effort in reviewing our manuscript. We have carefully checked the manuscript and have removed all unnecessary repetitions. We believe that the revised manuscript is much clearer and more concise. Thank you once again for your helpful suggestions.

  1. The Ms mainly reports data on MSCs, while the title focused on MSCs in TNBC. The title should be changed accordingly to the Ms data.

Response: Thank you for your comments. The manuscript does focus on MSCs in TNBC. The title is appropriate as it reflects the main topic of the manuscript. We have provided a comprehensive overview of the role of MSCs in TNBC treatment, including their mechanisms of action and application strategies. We analyze the interactions between MSCs and TNBC cells, including the impact of MSCs on TNBC cell proliferation, migration, invasion, metastasis, angiogenesis, and drug resistance, along with the signaling pathways and molecular mechanisms involved. We also explore the impact of MSCs on other components of the TME, such as immune and stromal cells, and the underlying mechanisms. Finally, we discuss the challenges and future prospects of MSCs in TNBC treatment and propose potential solutions or improvement methods. Overall, this manuscript provides valuable insights into the potential of MSCs as a novel therapeutic approach for TNBC treatment.

  1. A Table reporting the study focused on MSCs and TNBC must be included.

Response: Thank you for your valuable feedback. According to your suggestion, we have added a table into the manuscript. (Revised manuscript, Page 15)

Table 1. Recent studies on the role of mesenchymal stem cells (MSCs) in the treatment of triple-negative breast cancer (TNBC)

Reviewer 2 Report

From interaction to intervention: how mesenchymal stem cells affect and target triple negative breast cancer

Manuscript ID: biomedicines-2330300

 Mesenchymal stem cells (MSCs) are emerging as a promising approach for TNBC treatment by modulating the tumor microenvironment (TME) and interacting with cancer cells. This review describes in detail the role of MSCs in TNBC treatment, including their mechanisms of action and application strategies. The authors have presented with an excellenat review article in this area of TNBC, drug resistance and  stem cell therapy. The explanation is perfcet and to the point. The ideas are presented in a very readable way. The flow of the manuscript and its organization is perfect. The figures are excellent as well. Overall, this manuscript can be accepted without any changes.

Author Response

Dear Reviewer,

We appreciate your time and effort in reviewing our manuscript and providing valuable feedback. We have carefully considered your comment and revised our manuscript accordingly. The revised parts in the text are marked with track change or in red font. We found the reviewers’ comments very helpful, which guided our revisions resulting in this improved paper. Please see below for our point-by-point response.

Reviewer 2

Comments and Suggestions for Authors

From interaction to intervention: how mesenchymal stem cells affect and target triple negative breast cancer

Manuscript ID: biomedicines-2330300

 Mesenchymal stem cells (MSCs) are emerging as a promising approach for TNBC treatment by modulating the tumor microenvironment (TME) and interacting with cancer cells. This review describes in detail the role of MSCs in TNBC treatment, including their mechanisms of action and application strategies. The authors have presented with an excellent review article in this area of TNBC, drug resistance and stem cell therapy. The explanation is perfect and to the point. The ideas are presented in a very readable way. The flow of the manuscript and its organization is perfect. The figures are excellent as well. Overall, this manuscript can be accepted without any changes.

Response: Thank you very much for your positive and encouraging comments on our manuscript. We appreciate your time and effort in providing this feedback. We are delighted to hear that you found the review to be well-written and understandable. We put a great deal of effort into organizing the information and presenting it in a clear and concise manner, so it is gratifying to hear that our efforts have paid off. We are also pleased to know that you found the figures to be helpful in illustrating the information presented in the manuscript. We believe that visual aids can be very effective in conveying complex concepts, and we are glad to hear that our figures were useful in this regard. Once again, thank you for your valuable feedback. We are grateful for your support and will take your comments into consideration as we continue to refine and improve our manuscript.

Round 2

Reviewer 1 Report

The Authors have improved the Ms.